# Left dorsolateral prefrontal cortex supports context-dependent prioritisation of off-task thought

A. Turnbull [1], H.T. Wang [1], C. Murphy[1], N.S.P. Ho[1], X. Wang[1], M. Sormaz[1], T. Karapanagiotidis[1], R.M. Leech [2], B. Bernhardt[3], D.S. Margulies[4], D. Vatansever [5], E. Jefferies[1] & J. Smallwood[1]

When environments lack compelling goals, humans often let their minds wander to thoughts with greater personal relevance; however, we currently do not understand how this context-dependent prioritisation process operates. Dorsolateral prefrontal cortex (DLPFC) maintains goal representations in a context-dependent manner. Here, we show this region is involved in prioritising off-task thought in an analogous way. In a whole brain analysis we established that neural activity in DLPFC is high both when 'on-task' under demanding conditions and 'off-task' in a non-demanding task. Furthermore, individuals who increase off-task thought when external demands decrease, show lower correlation between neural signals linked to external tasks and lateral regions of the DMN within DLPFC, as well as less cortical grey matter in regions sensitive to these external task relevant signals. We conclude humans prioritise daydreaming when environmental demands decrease by aligning cognition with their personal goals using DLPFC.

[1] Department of Psychology, University of York, York, UK. [2] Centre for Neuroimaging Science, Kings College, London, UK. [3] Multimodal Imaging and Connectome Analysis Lab, Montreal Neurological Institute and Hospital, McGill University, Montreal, Quebec, Canada. [4] Centre National de la Recherche Scientifique (CNRS) UMR 7225, Institut du Cerveau et de la Moelle epiniere, Paris, France. [5] Institute of Science and Technology for Brain-inspired Intelligence, Fudan University, Shanghai, People's Republic of China. Correspondence and requests for materials should be addressed to A.T. (email: agt520@york.ac.uk)

Humans often use periods of low environmental demands to consider topics with greater relevance than events in the here-and-now[1,2]. Studies have linked the capacity to self-generate trains of thought that are decoupled from external input with beneficial psychological features, including delaying gratification[3], creative problem solving[4–7], and in refining personal goals[8]. Other studies, particularly those that measure ongoing experience in externally demanding task contexts, have shown that off-task self-generated thought has been linked to worse executive control and can be a cause of poor performance[9–13]. It has been argued that this apparent contradiction could be reconciled by assuming a general role of control processes that maximises the fit between patterns of ongoing experience and the demands imposed by the external environment[1]. This view, known as the context regulation hypothesis, predicts a common control process underpins both the act of reducing off-task thought when external demands are high, and increasing thoughts about personally relevant information when the environment lacks a compelling goal.

The context regulation hypothesis is hard to test behaviourally because studies have shown that periods of off-task experience interfere with task performance[9,14], suggesting that their occurrence can bias task-based estimates of an individual's working memory capacity[11,13]. Accordingly, this study addresses this gap in the literature by using covert measures of cognition derived from functional magnetic resonance imaging (fMRI) to understand how individuals prioritise off-task experience when task demands are low. Based on prior neuroimaging studies, the process of goal-motivated prioritisation may depend upon regions that make up the ventral attention, or salience, network[15]. This network includes regions of dorsolateral frontal (Brodmann Area, BA, 9/46) and parietal cortex (BA 40), the anterior cingulate (BA 24 and 32), as well as structures including the anterior insula. This network is important in influencing the maintenance of tasks sets[16] across a broad range of contexts, including listening to music[17], pain[18], and states of empathy/theory of mind[19]. The wide range of contexts within which the ventral attention network influences neural dynamics and cognition, suggests that it could be important in the process of context regulation. Consistent with this perspective, a previous study from our laboratory found that individual variation in the connectivity of the ventral attention network was related to population variation in the context-dependent regulation of off-task thought[20]. In two experiments combining measures of experience with neural function, we test the hypothesis that the ability to prioritise personally relevant thoughts during periods of low external demand depends on a domain-general neuro-cognitive process that helps aligns cognition with the most currently relevant goal[21]. In particular, we examine (a) whether a common neural region is involved in both the prioritisation of off-task thought when task demands are low and the facilitation of on-task thought when environmental demands are increased, and (b) the neural mechanisms that help individuals focus attention on personally relevant information under these circumstances.

## Results

**Identification of an off-task thought component**. To create conditions varying the requirement for external attention, we used a paradigm, which alternated between a higher demand condition in which task-relevant information is maintained in working memory (1-back) and a condition with no equivalent requirement (0-back, Fig. 1). While performing these tasks, participants intermittently provided descriptions of their ongoing thoughts using multidimensional experience sampling (MDES). This entails the participants describing their experience along a variety of questions, including whether they were thinking about the task, focused on themselves, or on future or past events (see Supplementary Table 1 for the full set of questions). In this paradigm, we routinely observe that participants engage in more off-task personally relevant thoughts in the easier 0-back paradigm[22,23]. In this study there were 24 MDES probes in the scanning experiment, yielding a total of 1438 observations for Experiment 1, and 30.7 on average in each session in the behavioural laboratory, yielding a total of 4482 observations in Experiment 2. We applied principal component analysis separately to the MDES data recorded in each dataset, in both cases identifying an off-task dimension (low-component loadings on-task, high loadings on episodic and social content). These components are presented as wordclouds in Fig. 1 and are highly similar across datasets ($r(11) = 0.882$, $p < 0.001$). Individual variation in off-task thinking was correlated across settings (0-back $r = 0.475$, $p = 0.002$; 1-back $r = 0.389$, $p = 0.014$) and more common in the 0-back task in both experiments (scanner: $t(59) = 5.997$, $p < 0.001$, lab: $t(145) = 7.120$, $p < 0.001$). In Experiment 1, neural data was acquired while participants performed this task and greater activity in superior parietal, sensorimotor, and mid-cingulate cortex was observed during 1-back blocks. Activity was greater in medial prefrontal, cingulate, and temporal cortex in 0-back blocks (Fig. 1 and Supplementary Table 2), replicating previous studies[23,24].

**fMRI analysis to find regions related to context regulation**. Having established patterns of off-task thinking using MDES, we next examined the neural associations with these patterns of thinking. In particular, we focused on how individuals prioritise this information when external demands are low, and how they prioritise on-task thinking when task demands are increased. We examined associations between momentary changes in off-task thinking and associated patterns of neural activity in both tasks (see Methods). If a neural region represents the prioritisation of cognition in line with external demands[21], stronger neural responses should occur in this region when focusing on (a) task-relevant information in a situation of increased task demand, and (b) personally relevant information in situations with reduced task demands. We performed a whole-brain fMRI analysis to see whether any regions of the brain had this neural profile (see Supplementary Table 2 for a full description of these results). We found that greater off-task thinking in the 0-back, and greater on-task in the 1-back task were associated with increased neural activity within left dorsolateral prefrontal cortex (DLPFC) (Fig. 2). To understand how our findings related to cognitive functions most commonly associated with these areas by prior studies in the literature, we performed a meta-analytic decoding using Neurosynth[25]. This analysis identifies terms in the literature most commonly associated with specific brain regions. Meta-analytic decoding of left DLPFC region identified through our analysis highlighted the term "executive" as most appropriate, indicative of a role for the region in cognitive control. To understand how this region fit into the broader neural architecture, we performed a seed-based functional connectivity analysis (see Supplementary Fig. 4). Intrinsic functional connectivity was observed with anterior insula, mid-cingulate cortex, anterior temporal parietal junction, regions that form the ventral attention network (VAN), which has been shown to play a role in task-set maintenance[16], attentional re-orienting, and contextual cueing[26].

We also found that bilateral clusters in the intraparietal sulcus (Fig. 2) were linked to more on-task thought across both tasks. Meta-analytic decoding revealed general task properties (e.g., "goal", "attention", "switching", "task") and more specific associations with external numeric tasks (e.g., "calculation").

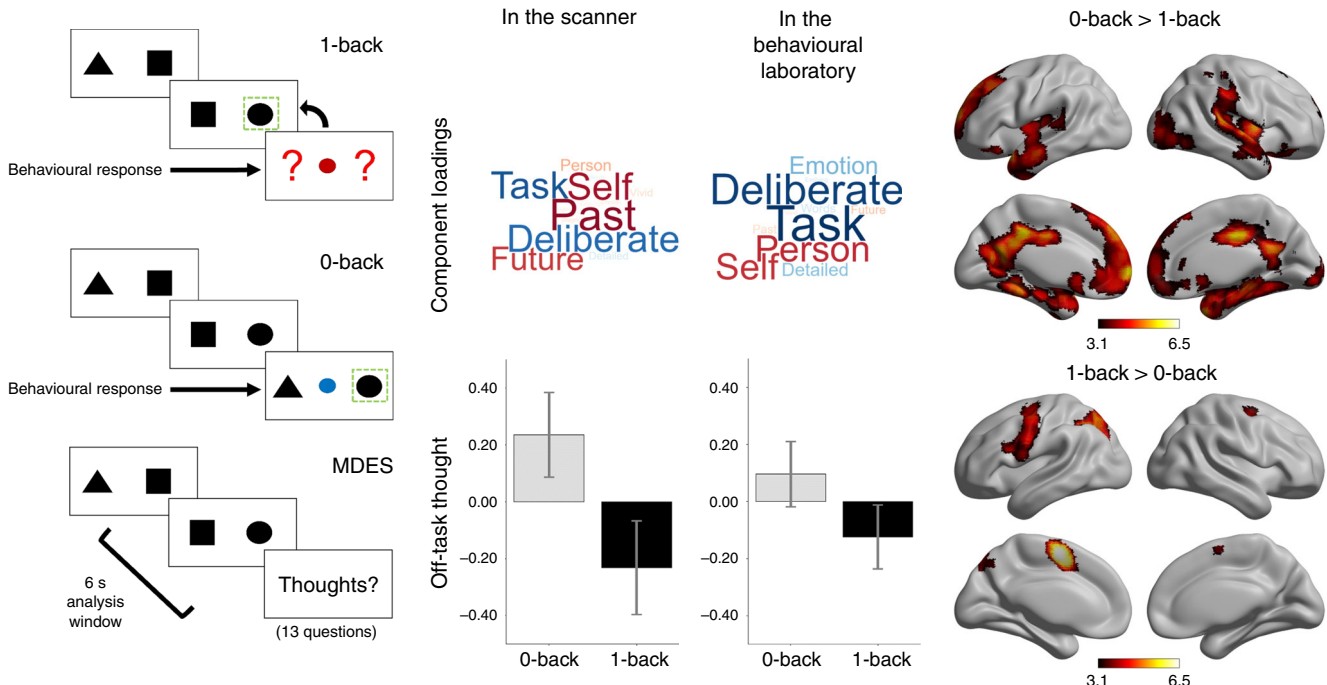

**Fig. 1** 0-back and 1-back tasks vary in their need for external attention in the scanner and the laboratory. Participants performed alternating blocks of two tasks (left). In the 0-back task, off-task thinking was increased (middle) in both the laboratory and scanner. The application of principal component analysis to MDES data identifies dimensions of thought by grouping questions that capture shared variance. One component identified in this manner captures a dimension that varies from a focus on the task to thoughts about the self and other and with an episodic focus, corresponding to one common definition of off-task mind-wandering[70]. The loadings on this component are presented in the form of wordclouds. Words in a larger font indicates items with a greater loading on the dimension and the colour describes the direction of this loading (red: positive; blue: negative). The average score for this off-task dimension of thought in each task is shown in the bar graphs in which the error bars indicate the 95% confidence intervals of the mean. Contrasts comparing neural activity across these conditions showed increased activity in default mode network regions during the 0-back, and left lateralised frontal and parietal regions during the 1-back (right). Task maps are corrected with a cluster-forming threshold of $Z > 3.1$, at a family-wise error rate of $p < 0.05$

Functional connectivity was observed with lateral frontal, mid-cingulate and temporo-parietal cortex, corresponding to the dorsal attention network (DAN). This network shows activity during spatial-orienting of visual attention and exerts top-down control over visual areas[26].

Experiment 1 establishes two aspects of the neural correlates of off-task thinking across situations with varying environmental demands. First, neural activity in left DLPFC is correlated with being on-task when task demands are higher, and off-task thoughts when demands are lower. This suggests that within left DLPFC periods of personally relevant concerns under situations of lower external demand share a similar neural correlate to periods of task-focused thought in a more demanding task context. Second, dorsal parietal cortex was associated with being on-task in both conditions, suggesting a more specialised role in external task-relevant processes in regions of the intraparietal sulcus, and a more abstract role in DLPFC that reflects the relationship between ongoing cognition and the level of external demands.

**Network interactions within DLPFC relate to off-task thought.** Experiment 1 identifies left DLPFC as showing a common neural profile whenever patterns of ongoing cognition match the demands of the environment. Studies in humans and monkeys suggest DLPFC monitors information in working memory[27,28] to form a zone of contextual control important for influencing information entering working memory[28,29]. Contemporary accounts of ongoing thought argue these experiences require a process of functional decoupling of neural signals related to self-generated information from signals, which directly reflect

environmental input[21]. Extrapolating from these accounts, we hypothesised that context-dependent variation in the association with off-task thought observed in our prior analysis occurs because of how neural signals related to the external task (i.e., posterior elements of the DAN) are processed in left DLPFC. In Experiment 2, we analysed resting-state and structural MRI data from 146 individuals who completed the same task in the behavioural laboratory, seeking evidence that neural processing within the left DLPFC is related to an individual's propensity for engaging in off-task thought when task demands are reduced. Unlike Experiment 1, this analysis examines off-task thinking from the perspective of a trait (see refs. [9–11,30,31] for prior examples of such an approach). Accordingly, sessions in Experiment 2 took place across three separate days to maximise the chances that our MDES captured a reasonably consistent description of the patterns of experience of each individual.

The pattern of association between activity in DLPFC and patterns of on-task/off-task thought observed in Experiment 1 could indicate that neural signals that reflect both task-related and self-generated information are processed within this region of cortex. To test this possibility in Experiment 2 we performed an analysis to determine (a) whether neural signals arising from other regions of cortex are observed in the DLPFC and (b) if the interaction between these signals explained population variation in context regulation. Following Leech et al.[32], we began our analysis by identifying how the timeseries of 17 well-established networks[33] are represented in left DLPFC, parcellating this region into partially overlapping sub-regions or "echoes"[32] corresponding to each network (see Methods). In the context of our experiment, these correspond to aspects of the left DLPFC in

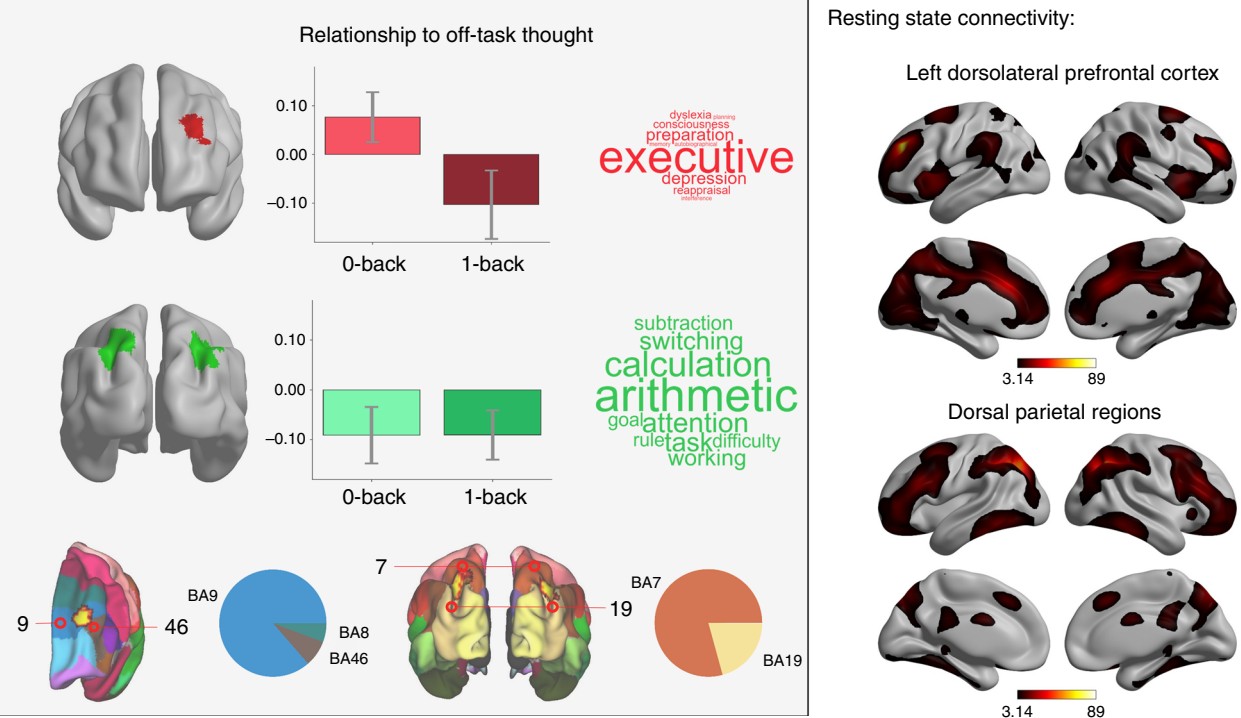

**Fig. 2** Establishing regions supporting on-task experience and those involved in the regulation of ongoing thought in line with the demands of the external environment. A region of dorsolateral prefrontal cortex (BA8, 9, and 46) was related to off-task thought during the 0-back and on-task thought during the 1-back (top and bottom left). Bilateral parietal regions (BA7 and 19) were related to on-task thought irrespective of task demands (middle left, centre bottom). The pie charts indicate the overlap of the regions identified by our analysis with Brodmann areas to enable a clearer understanding of their anatomical location. These regions show different patterns of resting-state functional connectivity (right). Wordclouds represent associations from meta-analytic decoding[25]. Statistical thresholds are identical to those in Fig. 1

which the observed neural signals within our region of interest are correlated with signals arising from other regions of cortex. Next, we produced a matrix of network interactions within DLPFC, which describes how correlated each of these signals was for each individual, allowing us to test how the functional coupling of signals from different networks predicts experience in the lab. Finally, this matrix was analysed to examine if they predicted individual variation in patterns of off-task thought recorded outside the scanner. We hypothesised that decoupling of signals related to external processing based on Experiment 1 (i.e., posterior regions of the DAN) should be linked to greater off-task thought in the 0-back task. Consistent with expectations based on Experiment 1, more off-task thoughts in the 0-back task was related to lower correlation/more negative correlation between Network 5, corresponding to posterior aspects of the DAN, and Network 17, lateral regions of the Default Mode Network (DMN) ($F(1,135) = 12.794$, $p = 0.0005$, see Fig. 3). No similar relationship was observed for off-task thought in 1-back blocks.

**Cortical thickness within DLPFC relates to off-task thought**. Our final analysis examined whether individual differences in a more stable neural trait was also related to elevations in off-task thought in the 0-back task by examining experiential associations with the grey-matter structure of the left DLPFC. Our functional analysis indicated signals arising from DAN had a complex topographic pattern within DLPFC, with positive coupling within a dorsal region (BA 9) and negative coupling in a ventral region (BA 46, Fig. 4). This separated the region along the border of a sulcus, with the more dorsal region coupled positively to signals related to the task, and the more ventral portion related negatively to the same signals. We hypothesised that if these regions play an

important functional role in how individuals focus on self-generated information, then increasing off-task thinking in the 0-back task should be linked to relatively less cortical thickness in regions of left DLPFC sensitive to signals from the DAN. Consistent with this view, relative reduced cortical thickness in dorsal relative to ventral regions was associated with greater off-task thoughts in less demanding conditions ($F(2,136) = 3.303$, $p = 0.040$; Fig. 4 and Methods).

**Selectivity of the left DLPFC to off-task thought**. To address the selectivity of the association between neural process in the DLPFC and on-task thought, we performed a number of post-hoc analyses. First, using the data we collected in Experiment 1 we extracted the relationship between brain activity in the same area of DLPFC and the other components of thought (Detail, Modality, and Emotion) to see if this region played a role in task-dependent regulation of these. We subtracted the relationship to each component in the 0-back from that in the 1-back, and used the effect seen for off-task thought in Experiment 1 (Cohen's $d = 0.48$) to define the size of the effect we were interested in. We performed equivalence tests[34] to see if the relationship between the task and thoughts for any other component could be dismissed as null. These were all significant (Detail: $t(59) = 1.978$, $p = 0.026$; Modality: $t(59) = 2.614$, $p = 0.006$; Emotion: $t(59) = 3.300$, $p = 0.001$), suggesting these effects are equivalent to zero and can be rejected as null effects (see Supplementary Fig. 5). We were significantly powered to perform this analysis (recommended $n = 38$, effect size Cohen's $d = 0.48$, alpha $= 0.05$, power $= 80\%$, two-one-sided $t$-tests (TOST) effect size calculation according to ref. [34])). This analysis indicates that task-relevant differences in the association with experience were only

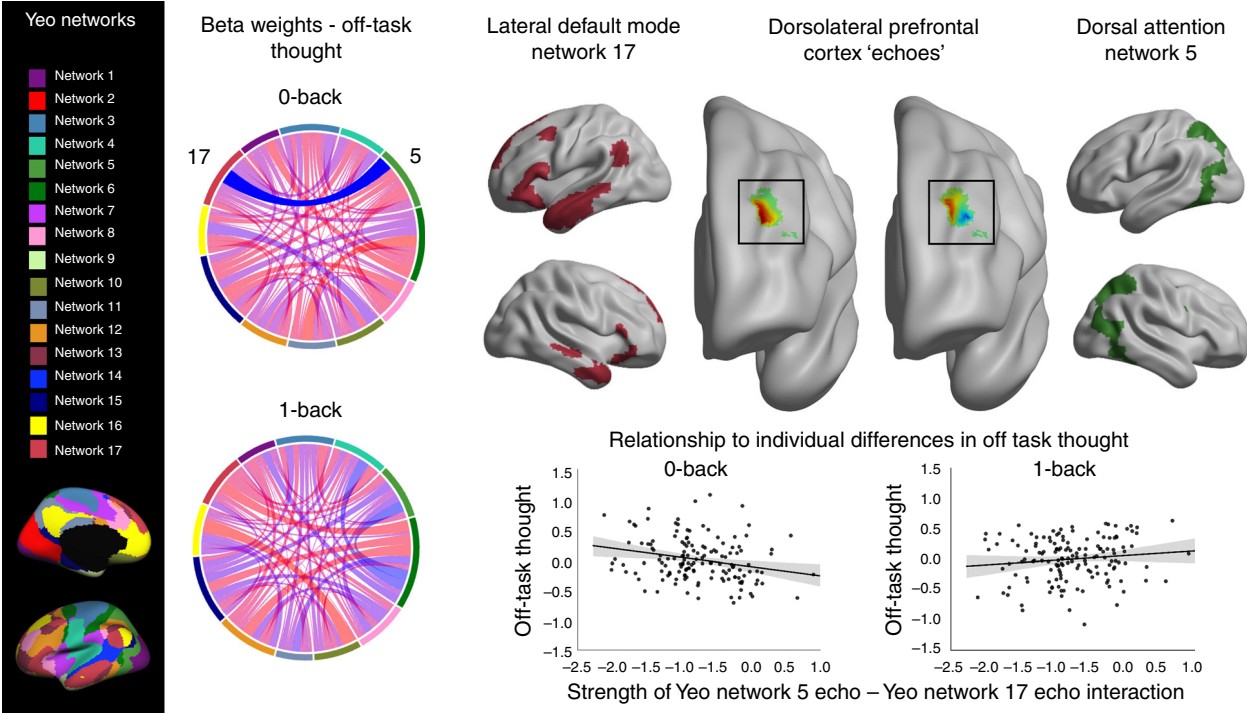

**Fig. 3** Segregation between echoes of the dorsal attention network and lateral temporal elements of the DMN relate to off-task thoughts in the 0-back condition. Analysis revealed a significant relationship between the correlation of Network 5 and 17 within left DLPFC and off-task thoughts during the 0-back. Chord diagrams represent beta-weights describing the relationship between the strength of pairs of network interactions and reports of thoughts in the 0-back and 1-back tasks. The significant relationship is highlighted (opaque). A key for the networks from Yeo et al.[33] is shown on the left, and the chord diagram colours correspond to these. A full description of these networks can be found in Supplementary Fig. 8

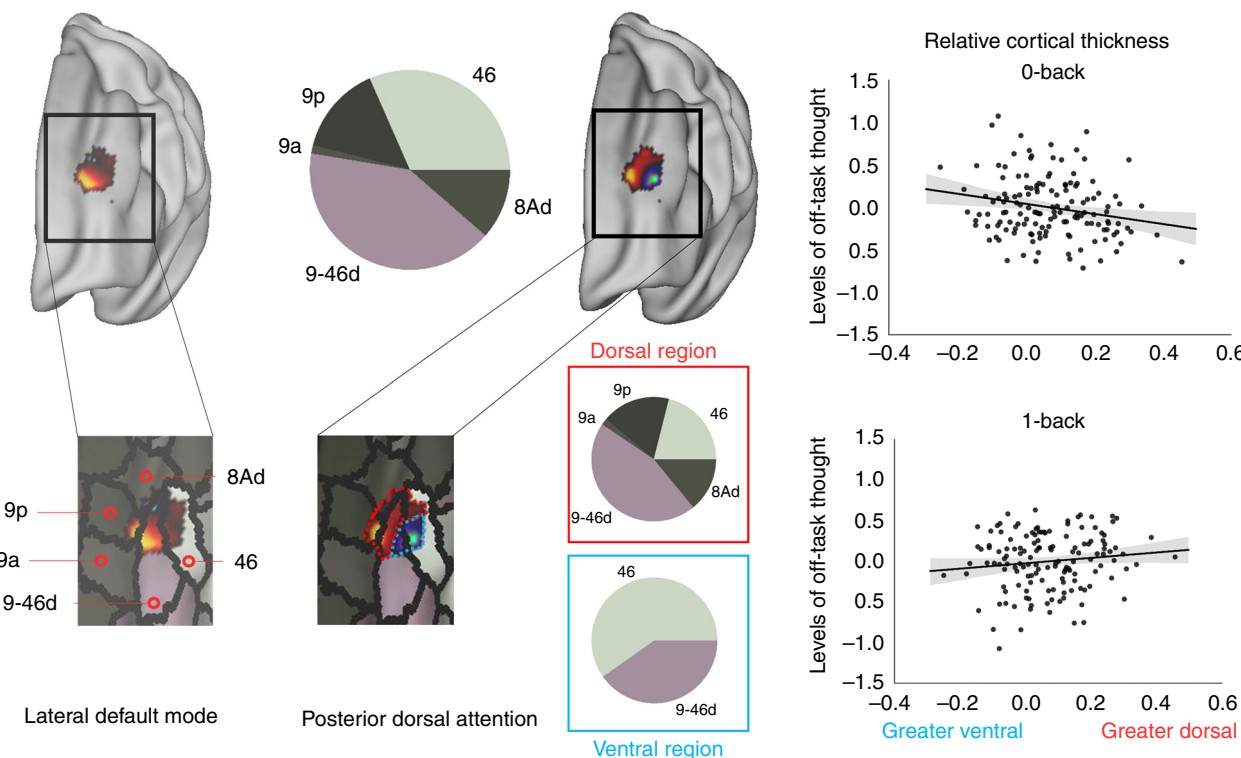

**Fig. 4** The structure of left DLPFC supports individual differences in prioritising off-task thought. Characterisation of the region of left DLPFC using a multimodal parcellation scheme[68] demonstrates it encompasses region BA 9/46d, 9, 46, and 8Ad. The amount of overlap with each parcel of the Glasser atlas[68] is shown by the pie charts, both for the region as a whole and for each sub-region (dorsal: red box, ventral: blue box). Relatively greater cortical thickness in a region negatively related to Yeo network 5[33] (posterior dorsal attention network) was linked to more off-task thought when task demands are lower. This relationship is shown in the scatterplots

significant within the left DLFPC for the off-task component of our MDES data. We also performed an equivalence analysis that examined how unique the associations are between cortex-wide signals and patterns of experience within left DLPFC that was observed in Experiment 2 (see Supplementary Table 3). We were significantly powered to perform this analysis (recommended $n = 97$, effect size $r = 0.2942$, alpha = 0.05, power = 80%, TOST effect size calculation). In brief, this found that no other pattern of experience could be predicted based on interactions between the same pair of networks (posterior DAN and lateral DMN) assuming we were looking for an effect of a statistically equivalent size to our significant finding. Moreover, of all the other network pairs included in our analyses all but one association with experience failed to pass Bonferroni correction for the number of comparisons. The outstanding pattern indicated associations between a different pair of networks (network 10, anterior limbic; network 16, DMN core) within DLPFC as related to the level of subjective detail in thoughts. Coupling between signals from these networks was associated with levels of detail in the 1-back task ($F(1,135) = 14.014$, $p = 0.0003$). The same equivalence analysis for this effect showed that the relationship between this interaction and detail in the 0-back was potentially of a comparable size and so could not be dismissed as a null finding. Additionally, the effect of this interaction on off-task thought in the DLPFC in both tasks was also too large to dismiss as definitely null. This means that the relationship between detailed thought in the 0-back, and task-related thought in both tasks, and the interaction between network 10 and network 16 within DLPFC was not statistically significant but was not significantly equivalent to 0, suggesting these relationships are too uncertain to draw firm conclusions. Post-hoc analysis showed that the correlation between these network components in DLPFC was positively related to detailed thought in the 0-back and negatively related to it in the 1-back (see Supplementary Fig. 7). Similarly, this same interaction was related negatively to off-task thought in the 0-back and positively in the 1-back. This suggests that while this region was not identified as related to detail during task performance, there may be signals in this region that also describe the task-relevant moderation of levels of detail, and it cannot be ruled out that these same signals relate to off-task thought.

Next, we repeated this analyses using bilateral parietal regions linked to on-task thought in Experiment 1 as the regions-of-interest (see Methods for explanation, Fig. 2 for the regions-of-interest, and Supplementary Table 3 for equivalence results). This found no comparable evidence that integration of distributed neural signals in these regions of parietal cortex are linked to patterns of experience. Finally, we repeated the whole-brain analysis from Experiment 1 looking at the neural correlates of the other components of experience identified by principal component analysis (PCA). This revealed one significant effect: reports of detailed thought were significantly more positively associated with neural signals in the posterior cingulate cortex in the harder 1-back task than in the easier 0-back task (see Supplementary Fig. 6). Taken together these supplementary analyses show that (i) off-task thought was the only pattern of experience that was associated with clear task differences in its association with neural activity in the left DLPFC during task performance (Experiment 1) and (ii) the association between signals from the posterior DAN and the lateral DMN within DLPFC only are specifically related to the prioritisation of personally relevant information when external demands are reduced (Experiment 2).

## Discussion

Our study combined multiple neuroimaging methods to demonstrate a role for left DLPFC in the prioritisation of personally relevant information in situations of low demands. To capture situations when individuals prioritise personally relevant thoughts when environmental demands are lower, we used a paradigm in which the low-demand condition was associated with greater off-task thought[8,22,23]. Experiment 1 found that within this context neural signals in left DLPFC were associated with off-task thought when task demands are lower, and on-task thought when demands are higher. Importantly, this pattern contrasted with neural signals within a parietal aspect of the DAN, which showed a positive association with on-task thought in both tasks. Examining neural processing within the left DLPFC, Experiment 2 found that the capacity of an individual to generate off-task thought in the low-demand condition was related to the degree of decoupling of neural signals arising from regions of posterior DAN, and involved in external task focus, from those from the lateral DMN. Further underlining the role of the DLPFC in off-task thought when environmental demands are reduced, we found that increases in cortical thickness in regions negatively related to task-relevant signals, relative to those positively linked to the posterior DAN, were linked to greater off-task thought. Altogether this pattern indicates that (a) under circumstances when off-task thought is high, periods of greater neural activity within the left DLPFC are linked to the emergence of increased personally relevant off-task thought and (b) that individuals who exhibit this capacity most clearly show a greater separation of functional signals between those linked to external task focus (the posterior DAN) and lateral regions of the DMN. Altogether, these provide converging support for the involvement of DLPFC in the process of prioritising cognition that matches the demands of a particular context.

One important implication of these findings is that they provide resolution to a long-standing debate within the literature on mind-wandering. It is currently unclear whether executive control suppresses[35], or facilitates off-task thinking[36], with behavioural evidence consistent with both perspectives[9,37–40]. Critically, behavioural studies alone may struggle to dissociate these positions because periods of off-task thinking during behavioural tasks measuring executive control are linked to poor performance[11,13]. Our neural evidence suggests that these are complementary[41], rather than contradictory accounts, since we found that focusing on a task, or imagining different people, times, and places, depends on shared neural processes in left DLPFC. Our individual difference analysis suggests DLPFC helps prioritise off-task thought via reductions in the processing of external task-relevant signals[14], a position supported by evidence that lesions to this region prevent patients ignoring external sensory input[42]. In mechanistic terms, therefore, our data suggests DLPFC may contribute to the decoupling of attention from external input that is thought to be necessary for efficient processing of self-generated information[21].

Studies from humans and non-human primates suggest DLPFC prioritises task-relevant information in a context-dependent manner[16,27–29]—monitoring signals from internal and external sources, emphasising those with greatest relevance to current goals[27,43]. Our study suggests humans have co-opted this process, allowing us to explicitly prioritise processes such as daydreaming, rather than less compelling events in the here-and-now. Although the ability to imagine different times and places is important[4], failure to appropriately suppress self-generated experiences causes problems in education[44], the workplace[37], and while driving[45]. Accordingly, managing when we let our minds wander requires cognition to be regulated in a context-dependent manner, and our study highlights left DLPFC is important in this process.

Before closing it is worth noting that as well as highlighting the left DLPFC in the process through which off-task thought is

**Table 1 Participant demographics for each experiment**

| Experiment | Task-based fMRI | Resting state fMRI | Cortical thickness MRI |
|---|---|---|---|
| Number of participants | 60 | 146 | 142 |
| Age (years) | $M = 20.17$, S.D. $= 2.22$ | $M = 20.21$, S.D. $= 2.49$ | $M = 2.23$, S.D. $= 2.47$ |
| Gender | 37 Female, 23 Male | 89 Female, 57 Male | 86 Female, 56 Male |

Thirty-nine participants performed both the resting state and task-based portions of this study. Cortical thickness analysis was performed in the same group as the resting-state analysis, but four participants were excluded as their structural data did not pass quality control

prioritised in a context appropriate manner, our data also implicate the DMN in how vivid and detailed experience is. Prior studies looking at population level variation in self-reports of the level of detail in patterns of ongoing thought show they are linked to neural processes within the DMN[30,46]. In this context, the current study provides both online evidence (see Supplementary Fig. 6) and individual difference analysis that complements our prior studies (see Supplementary Fig. 7). More generally, the view of the DMN as important for the level of detail in experience is consistent with prior studies that suggest details in memories are represented in posterior elements of the large scale network, including both the posterior cingulate[47] and angular gyrus[48,49]. Moreover, structural abnormalities in the posterior cingulate that emerge in dementia contribute to deficits in detail[50] and problems in generating a vivid scene in imagination[51]. Intriguingly, functional connections between the hippocampus and posterior cingulate cortex, which are associated with more detailed experiences in healthy individuals[30], is dysfunctional in dementia populations[52]. These observations provide converging evidence for a role of the DMN in features of how an experience is represented, such as its subjective detail. Altogether, this emerging literature provides the basis for a hypothesis of the contribution of the DMN in patterns of ongoing thought that future studies could explore. In particular, it will be helpful to use measures of neural function and experience across a wide range of situations to identify how broadly this relationship holds and identifying the causal role of DMN regions by studying populations with deficits within this system (such as Alzheimer's Disease) and by creating virtual lesions within this system in normal populations using techniques like transcranial stimulation[42].

Although our study provides important evidence implicating left DLPFC in the process through which we appropriately prioritise the nature of ongoing thought in a context-dependent manner, there are a number of important issues that remain unresolved. First, it is unclear whether the context-dependent nature of the role of left DLPFC in ongoing thought, conveys a behavioural advantage. Our study is unable to address this issue, in part, because although we found a consistent change in off-task thought across the 0-back and 1-back conditions in both experiments, we only observed a modulated pattern of behaviour in the larger behavioural study (see Supplementary Fig. 1). It is possible that this absence of a difference occurs because of the differences of the testing environment across the two experiments. Regardless of the reason for the absence of a behavioural difference in the scanner, in the future it will be important to determine whether left DLPFC is also important in facilitating behavioural efficiency across a range of different task contexts. Second, our study highlights left DLFPC as important in modulating ongoing thought across situations that on average vary in the degree to which they depend on continual focus on task-relevant information (Experiment 1) and that the degree to which individuals achieve this is related to neural patterns in the left DLPFC at rest (Experiment 2). In the future it will be important to use techniques that causally influence neural signals within this region (such as transcranial magnetic stimulation), or populations with lesions in this cortical region, to explicitly address whether this

region plays a causal role in how we exert control on our thoughts in order to ensure they are as aligned as possible with our goals.

## Methods

**Subject details.** See Table 1 for a full description of the sample in both experiments. In Experiment 1, 63 participants took part in the online task-based fMRI study. After excluding participants (see Method Details) 60 participants (37 females, mean age = 20.17 years, S.D = 2.22 years) remained for data analysis. Thirty-four participants from this sample were scanned for the data used in Sormaz et al.[22]. A group of 157 young adults were recruited for the resting-state fMRI and laboratory part of this study. After excluding participants (see Method Details) 146 participants remained for data analysis (89 females, mean age = 20.21 years, S.D = 2.49 years). These data have been used before by Turnbull et al.[20]. Of these participants, 39 also participated in Experiment 1. All participants were native English speakers, with normal/corrected vision, and no history of psychiatric or neurological illness. All participants were acquired from the undergraduate and post-graduate student body at the University of York. Both experiments were approved by the local ethics committee at both the York Neuroimaging Centre and the University of York's Psychology Department. All volunteers gave informed written consent and were compensated in either cash or course credit for their participation. A summary of the demographics can be seen in Table 1.

**Multidimensional experiential sampling (MDES).** Experience was sampled in a task paradigm that alternated between blocks of 0-back and 1-back in order to manipulate attentional demands and working memory load (Fig. 1). Non-target trials in both conditions were identical, consisting of black shapes (circles, squares, or triangles) separated by a line. In these trials the participant was not required to make a behavioural response. The shapes on either side of the line were always different. The colour of the centre line indicated to the participant the condition (0-back: blue, 1-back: red; mean presentation duration = 1050 ms, 200 ms jitter). The condition at the beginning of each session was counterbalanced across participants. Non-target trials were presented in runs of 2–8 trials (mean = 5) following which a target trial or multidimensional experience sampling (MDES) probe was presented.

During target trials, participants were required to make a behavioural response on the location of a specific shape. In the 0-back condition, on target trials, a pair of shapes were presented (as in the non-target trials), but the shapes were blue. Additionally, there was a small blue shape in the centre of the line down the middle of the screen. Participants were required to press a button to indicate which of the large shapes matched the central shape. This allowed participants to make perceptually guided decisions so that the non-targets in this condition do not require continuous monitoring. In the 1-back condition, the target trial consisted of two red question marks either side of the central line (in place of the shapes). There was a small shape in the centre of the screen as in the 0-back condition, but it was red. Participants had to indicate via button press which of the two shapes from the previous trial the central shape matched. Therefore, the decisions in this condition were guided by memory and so in this condition non-target trials had to be encoded to guide this decision.

The contents of ongoing thought during this paradigm were measured using multidimensional experience sampling (MDES). MDES probes occurred instead of a target trial on a quasi-random basis. When a probe occurred the participants were asked how much their thoughts were focused on the task, followed by 12 randomly shuffled questions about their thoughts (see Supplementary Table 1). All questions were rated on a scale of 1 to 4.

In the online task-based fMRI part of this study (Experiment 1), participants completed this task while undergoing fMRI scanning. Each run was 9-min in length and there were four runs per scanning session. In each run, there was an average of six thought probes (three in each condition), so that there were on average 24 (SD = 3.30, mean = 12 in each condition) MDES probes in each session. Two participants had one run dropped due to technical issues, leaving them with 18 MDES probes each.

In the behavioural laboratory (Experiment 2), to derive a reasonable stable estimate of each individual's patterns of thought, participants performed the task on 3 separate days in sessions that lasted around 25 min. In each session, the were eight blocks. In total, an average of 30.7 MDES probes occurred (SD = 5.7, mean = 15.4 in each condition). In the laboratory, accuracy was significantly greater ($t$ (145) = 9.487, $p < 0.001$) and reaction time significantly faster (t(145) = 14.362, $p <$

0.001) in the easier 0-back task. This effect was not found in either measure during fMRI scanning (accuracy: $t(59) = 0.369$, $p = 0.714$, rt: $t(59) = 0.052$, $p = 0.958$, see Supplementary Fig. 1).

**Resting-state (Experiment 2).** In the scanner, participants completed a 9-min eyes-open resting-state scan during which there was a fixation cross on-screen. Participants were instructed to look at the fixation cross and try to stay awake.

**fMRI acquisition.** All MRI scanning was carried out at the York Neuroimaging Centre. The scanning parameters were identical for both the resting state and online task-based scans. Structural and functional data were acquired using a 3T GE HDx Excite MRI scanner with an eight-channel phased array head coil tuned to 127.4 MHz. Structural MRI acquisition was based on a T1-weighted three-dimensional (3D) fast spoiled gradient echo sequence (TR = 7.8 s, TE = minimum full, flip angle = 20°, matrix size = 256 × 256, 176 slices, voxel size = 1.13 × 1.13 × 1 mm). Functional data were recorded using single-shot two-dimensional gradient echo planar imaging (TR = 3 s, TE = minimum full, flip angle = 90°, matrix size = 64 × 64, 60 slices, voxel size = 3 mm isotropic, 180 volumes). A FLAIR scan with the same orientation as the functional scans was collected to improve coregistration between scans.

**Data pre-processing: online task-based fMRI (Experiment 1).** Two participants were excluded for falling asleep. Task-based functional and structural data were pre-processed and analysed using FMRIB's Software Library (FSL version 4.1, http://fsl.fmrib.ox.ac.uk/fsl/fslwiki/FEAT/). Individual FLAIR and T1-weighted structural brain images were extracted using BET (Brain Extraction Tool). The functional data were pre-processed and analysed using the FMRI Expert Analysis Tool (FEAT). The individual subject analysis first involved motion correction using MCFLIRT and slice-timing correction using Fourier space timeseries phase-shifting. After coregistration to the structural images, individual functional images were linearly registered to the MNI-152 template using FMRIB's Linear Image Registration Tool (FLIRT). Functional images were spatial smoothed using a Gaussian kernel of FWHM 6 mm, underwent grand-mean intensity normalisation of the entire four-dimensional (4D) dataset by a single multiplicative factor, and both highpass temporal filtering (Gaussian-weighted least-squares straight line fitting, with sigma = 100 s); and Gaussian lowpass temporal filtering, with sigma = 2.8 s. An additional participant was excluded for having relative motion > 0.2 mm in >50% of runs (three participants total excluded).

**Data pre-processing: resting-state fMRI (Experiment 2).** Pre-processing of the resting-state fMRI data was carried out using the SPM software package (SPM Version 12.0, http://www.fil.ion.ucl.ac.uk/spm/) based on the MATLAB platform (Version 16.a, https://uk.mathworks.com/products/matlab.html). The individual subject analysis first involved motion correction with six degrees of freedom and slice-timing correction. Structural images were coregistered to the mean functional image via rigid-body transformation, segmented into grey/white matter and cerebrospinal fluid probability maps, and images were spatially normalised to the MNI-152 template. Functional images were spatially smoothed using an 8 mm Gaussian kernel; a slightly larger kernel was chosen to account for the increased sensitivity of functional connectivity analyses to signal-to-noise (SNR) issues. Owing to the additional problems associated with motion in functional connectivity analyses[53]; additional denoising procedures were carried out using the CONN functional connectivity toolbox (Version 17.f, https://www.nitrc.org/projects/conn[54]). An extensive motion correction procedure was carried out, comparable to that previously reported in the literature[55]. In additional to the removal of six realignment parameters and their second-order derivatives using a GLM[56], a linear detrending term was applied as well as the CompCor method with five principle components to remove signal from white matter and cerebrospinal fluid[57]. Volumes affected by motion were identified and scrubbed if motion exceeded 0.5 mm or global signal changes were larger than $z = 3$. Eleven participants that had >15% of their data affected by motion were excluded from the analysis[58]. Global signal regression was not used in this analysis due to its tendency to induce spurious anti-correlations[59,60]. A band-pass filter was used with thresholds of 0.009 and 0.08 Hz to focus on low-frequency fluctuations[61].

**Principal component analysis.** Behavioural analyses were carried out in SPSS (Version 24.0, 2016). The scores from the 13 experience sampling questions were entered into a PCA to describe the underlying structure of the participants' responses. Following prior studies[30,62] we concatenated the responses of each participant in each task into a single matrix and employed a PCA with varimax rotation. Four components were selected based on the inflection point in the scree plot (see Supplementary Fig. 3). These were defined as Task-relatedness, Detail, Modality, and Emotion of thought based on their question loadings. These loadings for both the scanner and laboratory components can be seen in Supplementary Fig. 2. Several analyses were performed to assess the similarity between PCA analyses in the laboratory and in the scanner. The PCA loadings for the off-task components were correlated across experimental conditions (Off-task: $r(11) = 0.882$, $p < 0.001$: Supplementary Fig. 2). The off-task PCA scores in each task condition were also correlated within the equivalent task condition for the 39 participants who took part in both parts of the experiment (Off-task: 0-back $r(37)$

= 0.475, $p = 0.002$, 1-back $r(37) = 0.389$, $p = 0.014$). Finally, paired $t$-tests were carried out to assess the differences in the off-task component between the 0-back and 1-back conditions (see results).

**Task-based fMRI analysis (Experiment 1).** Task-based analyses were carried out in FSL (FSL version 4.1, http://fsl.fmrib.ox.ac.uk/fsl/fslwiki/FEAT/). A model was set up for off-task thought by including four explanatory variables (EVs) as follows: EVs 1 and 2 modelled time periods in which participants completed the 0-back and 1-back task conditions; EVs 3 and 4 modelled the three thought probes in each condition, respectively, with a time period of 6 s prior to the MDES probe and the scores for the task-related component. This was convolved with the hemodynamic response function. We chose to use 6 s as it was the longest temporal interval within which no behavioural response occurred. Contrasts were included to assess brain activity that related to each task, as well as each component of thought. For the tasks, 0-back > 1-back and 1-back > 0-back contrasts were included. For the thoughts, main effects (positively or negatively related to thoughts in both conditions) and comparisons (activity related to thoughts in 0-back > thoughts in 1-back and vice versa) were included. The four runs were included in a fixed level analysis to average across the activity within an individual. Group level analyses were carried out using a cluster-forming threshold of $Z > 3.1$ and a whole-brain correction at $p < 0.05$ FWE-corrected. In these analyses we followed best practice as described by Eklund and colleagues[63]. Specifically, we used FLAME, as implemented in FSL, applied a cluster-forming threshold of $Z = 3.1$, and corrected these at $p < 0.05$ (corrected for family-wise error rate using random field theory). Average motion was included at the group level to additionally control for effects relating to this nuisance variable. This model was repeated for each component as a follow-up analysis (see below). Brain figures were made using BrainNet Viewer[64], plots made using matplotlib in Python (version 3.6.5). Meta-analytic decoding used Neurosynth[25] to find terms most commonly associated with our neural maps in the literature. This platform collects and synthesises results from many different research studies, and identifies the terms associated most often with each region of the brain.

**Resting state analysis (Experiment 2).** To understand whether the interaction of signals from the whole brain were implicated in the control of off-task thought within this region, we performed a modified version of an echoes analysis. In the original analysis[32], an independent component analysis was performed within a masked region of the brain, to identify different components within the region from its timeseries. These components represented voxels that grouped together in terms of their temporal signals, and they were shown to represent different functional networks within a single region of the brain. In our analysis, instead of identifying these components in a data-driven way, we used 17 well-established networks from the literature. The timecourse from these networks were correlated with the timecourse within the left DLPFC to identify components within this region that represented each network. To do this, the 17 Yeo network masks[33] were binarised and merged into a single 4D nifti file. In order to reduce statistical bias from the region itself, the corresponding region (left DLPFC) was masked out of this nifti. These networks were entered into a dual regression that extracted the timeseries from within each Yeo network and subsequently regressed these against each subjects 4D dataset within the left DLPFC. Thus, for each network, each voxel in the left DLPFC was given a value for how much it represented the Yeo network in question, to make up 17 different components within this region that each represented one of the Yeo networks. These maps were again merged into a single file and entered into the first step of a second dual regression to extract the timeseries of each Yeo 17 echo or component. FSL Nets (v0.6) was used to extract these timeseries and produce a matrix of interactions defined by the partial correlation between each set of echoes. These interactions were entered into a model as dependent variables to model their relationship to the average component scores (e.g., task-relatedness) in each task from the laboratory. We inspected each component, and if no voxels were significantly related to the Yeo network in question it was deemed to likely represent noise and its interactions were excluded from the analysis. An alpha value of $p < 0.05/17$ was used to account for family-wise error in this analysis. The average scores from each task were included as independent variables in the model (MANOVA), as well as the interaction between the scores in each task, and age, gender, and mean motion in order to additionally control for the effect of these covariates of no interest. This was used to identify any specific network interactions that could be predicted by the thoughts in either task. These results were Bonferroni corrected for the number of interactions in the model.

Several analyses were performed to assess the specificity of this result. First, we repeated this masked network interaction analysis with the other components of thought (Detail, Modality, and Emotion: see Supplementary Fig. 2), and found only one significant effect within the model for Detailed thought that passed Bonferroni correction, with the interaction between Yeo networks 10 and 16 within the DLPFC region-of-interest significantly related to the level of Detail in participants' thoughts in the 1-back task ($F(1,135) = 12.792$, $p = 0.000484$). Next, we repeated the analysis from Experiment 1 using the other components in turn (i.e., the PCA scores from Detail, Modality, and Emotion rather than Off-task). This revealed a region of posterior cingulate cortex whose neural signature was consistent with a task-dependent association with the level of detail in ongoing experience, in a similar way to the DLPFC was related to off-task thought (see Supplementary Fig. 6). To see whether it regulated detail using a similar mechanism, we repeated

the masked connectivity analysis described above on this region and there was no individual interaction that passed Bonferroni correction. No other components of experience showed a brain region with a pattern consistent with context regulation, but there were several results related to levels of thought in a non-task-dependent manner (see Supplementary Fig. 9). Third, we repeated the connectivity analyses from Experiment 2 using the whole brain, rather than limiting the analysis to the DLPFC. This analysis found no significant relationships between network interactions and off-task thought in any conditions, suggesting the association between the interaction of the dorsal attention network and the default mode network with off-task thought is not a property of the broader cortical mantle. Fourth, we explored whether this same analysis would identify similar effects in regions important for on-task thought regardless of the task (lateral parietal regions: see Fig. 2). This analysis revealed no significant relationships that passed Bonferroni correction, suggesting that the relationships between network interactions in DLPFC and off-task thought identified in our prior analyses were unique to regions that responded in a manner that was task-dependent.

Finally, to test whether the specific relationships we found (between the network 5-network 17 interaction and off-task thought, and the network 10-network 16 interaction and detailed thought) were truly unique to these conditions, we performed a series of equivalence analyses. These were done using the TOST equivalence test for correlations described in Lakens (2017)[34]. We extracted the strength of the relationship in our significant result (between the interaction of network 5 and 17 and off-task thoughts in the 0-back, and between the interaction of network 10 and 16 and detailed thought in the 1-back) and used these as the upper and lower bounds to see if there were any effects of this size under the other conditions. All of these tests using the network 5-network 17 interaction were significant, suggesting the correlations between this interaction and the thoughts under other conditions were equivalent to 0 and represent true null findings (see Supplementary Table 3). This suggests that this interaction is specifically able to predict off-task thoughts in the 0-back task within DLPFC. The network 10-network 16 interaction was able to significantly predict detail within the DLPFC in the 1-back task. An equivalence analysis suggested that the effect in the 0-back was of equivalent size and cannot be dismissed as a null effect. Interestingly, this also involves the default mode network, and a post-hoc analysis showed that the relationship to detail was task-dependent (see Supplementary Fig. 7). The relationship between this interaction and off-task thought also was not significant using this equivalence test, so this relationship cannot be dismissed as null. Post-hoc analyses showed that this relationship was also task-dependent (see Supplementary Fig. 7).

All data were mean centred before we performed this analysis. Chord diagrams were made using R: these represent the strength of the relationship between each interaction and the thoughts. Parameter estimates were extracted from the MANOVA so that each interaction had a beta-weight representing how strongly it related to the thoughts as part of the model. These were used to create chord diagrams that show the strength of these relationships (in the size of the chords) and their direction (blue is negative, red is positive).

**Cortical thickness analysis.** After identifying that the DAN echo appeared to span multiple regions, we performed a follow-up structural analysis that looked at whether the difference in cortical thickness between DAN-negative and DAN-positive regions also related to the levels of off-task thoughts. FreeSurfer was used to estimate vertex-wise cortical thickness (5.3.0; https://surfer.nmr.mgh.harvard.edu), using an automated surface reconstruction scheme described in detail elsewhere[65–67]. The following processing steps were applied: intensity normalisation, removal of non-brain tissue, tissue classification and surface extraction. Cortical surfaces were visually inspected and corrected if necessary. Cortical thickness was calculated as the closest distance between the grey/white matter boundary and pial surface at each vertex across the entire cortex. A surface-based smoothing with a full-width at half maximum (FWHM) = 20 mm was applied. Surface alignment based on curvature to an average spherical representation, fsaverage5, was used to improve correspondence of measurement locations among subjects. One-hundred forty-two of the 146 participants used in the previous analysis had cortical thickness extracted in a way that passed visual quality control. The scores for the DAN-negative region were subtracted from the DAN-positive region to give a difference score. This was entered into a MANOVA, with the off-task scores in each task as the dependent variables, as well as age and gender as covariates of no interest. All data were mean centred before this analysis. Overlaps were calculated and displayed with Brodmann and Glasser[68] labels using Connectome Workbench, with labels being acquired from the BALSA database[69].

**Reporting summary.** Further information on research design is available in the Nature Research Reporting Summary linked to this article.

## Data availability

Raw Z-maps from the task-based analysis are available on Neurovault in a collection with the title of this article, along with the significant components from the resting state and cortical thickness analyses. All summary data and materials used in the analysis are available on request. Raw fMRI data is restricted in accordance with ERC and EU regulations.

## Code availability

All code used in the production of this manuscript is available on request.

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

## Acknowledgements

Thanks to York Neuroimaging Centre for technical support. This project was supported by European Research Council Consolidator awarded to JS (WANDERINGMINDS–646927).

## Author contributions

A.T. and J.S. conceived of the ideas for analysis. Computations were performed by A.T. with assistance from H.T.W., R.M.L., M.S., C.M., T.K., and D.V. (functional data); B.B., N.S.P.H., X.W. and D.S.M. (structural data). H.T. W. and J.S. designed the task. The manuscript was written by A.T. and J.S. with contributions from B.B., E.J., R.M.L., T.K., D.M. and D.V.

## Additional information

**Competing interests:** The authors declare no competing interests.

