## [Peer Review File · Nature Communications]

Reviewers' Comments:

Reviewer #1:

Remarks to the Author:

Summary:

This paper is about the role of the DLPFC in mind-wandering and on-task thought. Using data from a large number of participants (n=60), they showed that DLPFC increases in activity for mind-wandering during a very easy task, while it increases in activity for being on-task during a slightly more difficult task. They then relate this to connectivity networks as well.

Main points:

I think the premise of the paper, that the DLPFC shows activity consistent with both mind-wandering and being on-task, is very interesting. However, at this time I have a hard time at understanding many things about the study. Moreover, I think some conclusions are a bit overconfident.

- p.3 It would be helpful to indicate the tasks: what did participants have to do? And how often were thought probes inserted? In addition about the task, it seems worrisome that there are no differences between the two task conditions in the scanner. it would be important to discuss that in the discussion.
- p. 5 It is not clear what meta-analytic decoding is and what research question it seeks to answer.
- Fig.1 It is not clear what the word clouds indicate and neither are the bar graphs under the word clouds. It would be helpful to have an interpretation guide.
- Fig.2 It is not clear what the pie charts indicate.
- p. 6 I don't think that the conclusion "Experiment 1 establishes two aspects of how individuals prioritize off-task thought when environmental demands are low" is warranted. This confounds brain activity with behavior: you have only demonstrated differences in brain activity, nothing about changes in prioritization of thought. For one thing, then you'd need to show there is a larger amount of off-task thought in the low-demand condition. Moreover, you would need to show that for people who prioritize off-task thought more, there is more DLPFC in the low-demand task, but less DLPFC in the high-demand task. When you have shown both, this gives some indication of the behavior you are seeking to describe.
- p.7 It is not clear what the "echoes" analysis is.
- p. 9 The segregation analysis is also a bit unclear. First of all, this relies strongly on the premise that whatever differences are measured between the two tasks are a stable individual difference, not dependent on circumstances, since you relate it to differences in brain structure, which obviously only show a person's long-term traits. Secondly, it is not quite clear why you expect the observed differences between the dorsal and ventral DLPFC. It would be helpful to explain the logic here. Finally, again the pie charts in Fig 4 are unclear.
- p.10 Refers to a debate in the mind-wandering literature--it would be helpful to identify this debate.
- p.10 Given the considerations above, I would phrase my conclusion less strongly: the DLPFC may help to prioritise cognition
- p.18 "This revealed a region of posterior cingulate cortex whose neural signature was consistent with a pattern of context regulation for the level of detail in ongoing experience" What is context regulation? How is this pattern consistent with that?
- p.19 Several null relationships are mentioned, but to be sure there are really no relationships (and this is not an artifact of data that are too noisy) you need to either compute Bayes Factors or perform equivalence analyses.

Minor points:

- p.2 suggested insertion: "We established neural activity in DLPFC is high BOTH when 'on-task' under demanding conditions and 'off-task' in a non-demanding task"
- p.2 "segregation": what is segregated from what?

- p.7 "Post-hoc analyses confirmed this pattern was absent from regions linked to on-task thought in Experiment 1" It would be helpful to give some examples of those regions here.
- Fig.3 This figure would be a lot easier to understand if the Yeo et al networks were associated with names.
- p.10 "Our individual difference analysis suggests DLPFC helps prioritise off-task thought via reductions in the processing of external task-relevant signals[3]," I find this very helpful. It would have been nice to explain this reasoning in the results already so it is clearer why the connectivity analyses were done.
- p. 16 "eleven participants excluded in total." seems to be an orphan sentence. Something like "leading to eleven participants being excluded in total." would make more sense.
- p.16 "The PCA loadings for the equivalent components were correlated across experimental conditions" It would be helpful to mention the actual correlation coefficients here.
- p.17 "with a time period of 6 seconds prior to the MDES probe and the scores for the task-related component" Did you take into account the delay in the HRF?
- p.17 The Yeo analysis is very hard to understand. Maybe it would be clearer if the purpose of each analysis step could be explained
- p.18 What is "voxel-wise correction at $p < .05/17$ "? What is the manova in the same paragraph trying to predict?
- p. 18 "we repeated this masked network interaction analysis with the other components of thought" Would be nice to give examples of these other components of thought.
- p.19 "Chord diagrams were made using R[40]." It would be nice to give some more explanation of the statistics underlying these graphs.
- Several references are missing page numbers and issue numbers. There is some inconsistency in the naming of journal names (such as PNAS).

Reviewer #2:

Remarks to the Author:

The paper presents very interesting findings about the DLPFC and its function in prioritizes goal representations in an adaptive, context-dependent manner. The two experiments presented are an impressive body of work and I think the paper could have a positive and meaningful impact on the field. However, I do have a few concerns that I think should be addressed before recommending publication.

A prominent issue is the lack of an Introduction that integrates related prior work and theoretical positions. This is critical because the paper presents results on relatively specific hypothesis, so it's important to understand where it came from and what alternative explanations may exist. A more thorough explanation of why they chose to focus on the left DLPFC would be especially helpful -- theoretically as well as related to previous research. Including this will assist readers/reviewers situate and evaluate the paper. (As an aside, I noticed that most of the articles published in Nature Communications do include short introductions.)

Another issue in general is the connection among the different analyses adopted in the paper. They are undoubtedly important, novel, and interesting. However, I think more could be said in the paper about how they naturally flow together, and what question each answers. An Introduction may also help alleviate this issue.

Some of the decisions that were made/analyses completed along the way are not described is a

weakness of the paper. For example, there is no mention of why the authors chose to focus on the left DLPFC in particular (rather than bilateral or right)? Why was window of 6 seconds chosen to analyze prior to the MDES questions? What is the significance/relationship of having the laboratory task as well as the scanner? Some of these questions and decisions likely have very good rationales, but they are left out of the paper. I realize there are significant space constraints, but I think some the information is necessary to include.

The discussion section is also very short. Although there are some summary statements in the paper, but I think it might be helpful to again link the findings to a broader context and describing how each main finding is important. Sometimes I was left wondering what questions a particular analysis answered, and how that fit with current theories/findings.

It would be helpful if the discussion section broadened a bit by exploring alternative explanations as well. How do you know if it is an actual prioritization mechanism, compared to another explanation? What other explanations may exist?

Why is the family wise error rate .05 (Fig 1)? In general, how were multiple comparisons dealt with here?

Although I think the Figures are very well-done and helpful, they could still use some improvement to assist with reader interpretation by including more labels and details in the figure and captions.

There may be a better name for the section "classification of off-task thought component." The classification part in particular was a bit confusing at first.

The writing quality in the Methods section could improve quite a bit. The other sections of the paper are written much more clearly/concisely in comparison.

The human subjects section is somewhat difficult to understand in terms of which participants came from what group/study. Perhaps a table would help display this information.

It would also be helpful to detail the number of instances for each measure (e.g. MDES in the scanner, lab, etc.). How many of these instances were used to create the PCA. Descriptive statistics may also be useful (in supplementary materials).

"Two participants were excluded for falling asleep" appears in more than one section. Are these the same two people?

Reviewer #3:

Remarks to the Author:

This is an exciting and potentially paradigm-shifting study on the brain basis of mind wandering. The experiment was cleverly-designed and well-executed; the analyses were rigorous and innovative, leveraging multiple cutting-edge approaches to fMRI data analysis to provide converging evidence in support of conclusions. The findings have important implications not only for research on mind wandering but also for other modes of self-generated thought as well as more general research on human brain networks and cognitive neuroscience.

I am very familiar with the work of this research group. In my view this study is among the most (if not the most) important to come from their lab. I do not recall ever recommending acceptance of a

paper without any changes, but I could not detect anything that should be changed about this paper, aside from the minor point below. I would like to congratulate the authors on an outstanding contribution to science.

Minor point

In the first sentence of the Discussion (line 2), I believe the authors meant to include the word "relevant" between "personally" and "information."

Response to reviews – Turnbull and colleagues

Reviewer #1 (Remarks to the Author):

Summary:

This paper is about the role of the DLPFC in mind-wandering and on-task thought. Using data from a large number of participants (n=60), they showed that DLPFC increases in activity for mind-wandering during a very easy task, while it increases in activity for being on-task during a slightly more difficult task. They then relate this to connectivity networks as well.

Main points:

I think the premise of the paper, that the DLPFC shows activity consistent with both mind-wandering and being on-task, is very interesting. However, at this time I have a hard time at understanding many things about the study. Moreover, I think some conclusions are a bit overconfident.

Thank you for your interest in our papers' key findings.

- p.3 It would be helpful to indicate the tasks: what did participants have to do? And how often were thought probes inserted? In addition, about the task, it seems worrisome that there are no differences between the two task conditions in the scanner. it would be important to discuss that in the discussion.

Thanks for pointing this out. In this revision we now provide a thorough description of the methods of the tasks, and explicitly consider the lack of a consistent behavioural difference in the scanner data in the discussion.

“Experience was sampled in a task paradigm that alternated between blocks of 0-back and 1-back in order to manipulate attentional demands and working memory load (Figure 1). Non-target trials in both conditions were identical, consisting of black

shapes (circles, squares, or triangles) separated by a line. In these trials the participant was not required to make a behavioural response. The shapes on either side of the line were always different. The colour of the centre line indicated to the participant the condition (0-back: blue, 1-back: red; mean presentation duration=1050ms, 200ms jitter). The condition at the beginning of each session was counterbalanced across participants. Non-target trials were presented in runs of 2-8 trials (mean = 5) following which a target trial or multidimensional experience sampling (MDES) probe was presented.

During target trials, participants were required to make a behavioural response on the location of a specific shape. In the 0-back condition, on target trials, a pair of shapes were presented (as in the non-target trials), but the shapes were blue. Additionally, there was a small blue shape in the centre of the line down the middle of the screen. Participants were required to press a button to indicate which of the large shapes matched the central shape. This allowed participants to make perceptually-guided decisions so that this condition does not require continuous monitoring. In the 1-back condition, the target trial consisted of two red question marks either side of the central line (in place of the shapes). There was a small shape in the centre of the screen as in the 0-back condition, but it was red. Participants had to indicate via button press which of the two shapes from the previous trial the central shape matched. Therefore, the decisions in this condition were guided by memory and this part of the task required constant monitoring in case each non-target trial had to be used to guide this decision.

The contents of ongoing thought during this paradigm were measured using Multi Dimensional Experience Sampling (MDES). MDES probes occurred instead of a target trial on a quasi-random basis. When a probe occurred the participants were asked how much their thoughts were focused on the task, followed by 12 questions about other features of experience (see Supplementary Table 1) presented in a random order. All questions were rated on a scale of 1 to 4.

In the online task-based fMRI part of this study (Experiment 1), participants completed this task while undergoing fMRI scanning. Each run was 9-minutes in length and there were four runs per scanning session. In each run, there was an average of six thought probes (three in each condition), so that there were on average 24 (SD=3.30, mean=12 in each condition) MDES probes in each session. Two participants had one run dropped due to technical issues, leaving them with 9 MDES probes each in each condition.

In the behavioural laboratory (Experiment 2), the task was performed on three separate days in sessions that lasted around 25 minutes, and these were separated into eight blocks. In total, an average of 30.7 MDES probes occurred (SD=5.7, mean=15.4 in each condition). In the laboratory, accuracy was significantly greater ($t(145)=9.487$, $p<.001$) and reaction time significantly faster ($t(145)=14.362$, $p<.001$) in the easier 0-back task. This effect was not found in either measure during fMRI scanning (accuracy: $t(59)=0.369$, $p=.714$, rt: $t(59)=0.052$, $p=.958$, see Supplementary Figure 1)."

"First, it is unclear whether the context dependent nature of the role of left DLPFC in ongoing thought, conveys a behavioural advantage. Our study is unable to address this issue, in part, because although we found a consistent change in off-task across the 0-back and 1-back conditions in both experiments, we only observed a modulated pattern of behaviour in the larger behavioural study. It is possible that this absence of a difference occurs because of the differences of the testing environment across the two experiments. Regardless of the reason for the absence of a behavioural difference in the scanner, in the future it will be important to determine whether left DLPFC is also important in facilitating behavioural efficiency across a range of different task contexts."

- p. 5 It is not clear what meta-analytic decoding is and what research question it seeks to answer.

We apologise for our lack of clarity on this aspect of our study. Neurosynth is a platform that collects and synthesises results from many different research studies.

It extracts coordinates from published papers and associates them with “terms of interest” to give the most commonly associated cognitive terms with each region of the brain. In our study, we used Neurosynth to highlight the cognitive functions most commonly associated with each of our regions of interest in the wider neuroimaging literature. We have added the following explanation to make this clearer to a reader:

“To understand how our findings related to the cognitive functions most commonly associated with these areas in the literature, we performed a meta-analytic decoding using Neurosynth (Yarkoni et al., 2011). This program identifies terms in the literature most commonly associated with specific brain regions.”

“Meta-analytic decoding used Neurosynth[19] to find terms most commonly associated with our neural maps in the literature. This platform collects and synthesises results from many different research studies, and identifies the terms associated most often with each region of the brain.”

- Fig.1 It is not clear what the word clouds indicate and neither are the bar graphs under the word clouds. It would be helpful to have an interpretation guide.

Thank you for pointing out this lack of clarity. We have added a more thorough explanation of these plots in this revision:

“The application of principal component analysis to MDES data identifies dimensions of thought by grouping questions that capture shared variance. One component identified in this manner captures a dimension that varies from a focus on the task to thoughts about the self and other and with an episodic focus, corresponding to one common definition of off-task mind-wandering[18]. The loadings on this component are presented in the form of wordclouds. Words in a larger font indicates items with a greater loading on the dimension and the the colour describes the direction of this loading (red: positive, blue: negative).”

- Fig.2 It is not clear what the pie charts indicate.

We thank the reviewer for highlighting this omission. We have added the following explanation:

“The pie charts indicate the overlap of the regions identified by our analysis with Brodmann areas to enable a clearer understanding of their anatomical location.”

- p. 6 I don't think that the conclusion "Experiment 1 establishes two aspects of how individuals prioritize off-task thought when environmental demands are low" is warranted. This confounds brain activity with behavior: you have only demonstrated differences in brain activity, nothing about changes in prioritization of thought. For one thing, then you'd need to show there is a larger amount of off-task thought in the low-demand condition. Moreover, you would need to show that for people who prioritize off-task thought more, there is more DLPFC in the low-demand task, but less DLPFC in the high-demand task. When you have shown both, this gives some indication of the behavior you are seeking to describe.

Thanks for this suggestion. We have moderated the conclusions in this paragraph. In this revision we now state:

“First, neural activity in left DLPFC is correlated with being on-task when task demands are higher, and off-task thoughts when demands are lower. This suggests that within the left DLPFC periods of personally-relevant concerns under situations of lower external demand share a similar neural correlate to periods of task-focused thought in a more demanding task context. Second, dorsal parietal cortex was associated with being on-task in both conditions, suggesting a more specialised role in external task-relevant processes in regions of IPS, and a more abstract role in DLPFC that reflects the relationship between ongoing cognition and the level of external demands.”

Also, please note that in our paradigm we do repeatedly find that off task thinking is higher in the 0-back task (see Experiment 1 and 2, as well as multiple other studies

using the same design – Smallwood et al., 2013, Medea et al., 2016; Konishi et al., 2017). Moreover, the results of Experiment 2 show that the degree to which individuals generate off task thoughts in the 0 back task is specifically related to the observed correlation between the posterior DAN (the region that supports external task focus) and the lateral DMN. Thus while we agree that the results of Experiment 1, in isolation, does not provide evidence of prioritisation, the combination of our two Experiments are closer to satisfying the requirements for constituting a role in the contextual prioritisation of off-task thought. We have attempted to be more explicit about this in the discussion of our revision.

“Our study combined multiple neuroimaging methods to demonstrate a role for left DLPFC in the prioritisation of personally relevant information in situations of low demands. To capture situations when individuals prioritise personally relevant thoughts when environmental demands are lower, we used a paradigm in which the low demand condition was associated with greater off-task thought [7, 15, 16]. Experiment 1 found that within this context neural signals in left DLPFC were associated with off-task thought when task demands are lower, and on-task thought when demands are higher. Importantly, this pattern contrasted with neural signals within a parietal aspect of the DAN, which showed a positive association with on-task thought in both tasks. Examining neural processing within the left DLPFC, Experiment 2 found that the capacity of an individual to generate off-task thought in the low demand condition was related to the degree of decoupling of neural signals arising from regions of posterior DAN, and involved in external task focus, from those from the lateral DMN. Further underlining the role of the DLPFC in off-task thought when environmental demands are reduced, we found that increases in cortical thickness in regions negatively related to task-relevant signals, relative to those positively linked to the posterior DAN, were linked to greater off-task thought. Together this pattern indicates that (a) under circumstances when off-task thought is high, periods of greater neural activity within the left DLPFC is linked to the emergence of increased personally relevant off-task thought and (b) that individuals who exhibit this capacity most clearly show a greater separation of functional signals between those linked to external task focus (the posterior DAN) and lateral regions

of the DMN. Together these provide converging support for the involvement of DLPFC in the process of prioritising cognition that matches the demands of a particular context.”

And we also explicitly consider this issue in the limitations section of this revision:

“Second, our study highlights left DLPFC as important in modulating ongoing thought across situations that on average vary in the degree to which they depend on continual focus on task relevant information (Experiment 1) and that the degree to which individuals achieve this is related to neural patterns in the left DLPFC at rest (Experiment 2). In the future it will be important to use techniques that causally influence neural signals within this region (such as transcranial magnetic stimulation), or populations with lesions in this cortical region, to explicitly address whether this region plays a causal role in how we exert control on our thoughts in order to ensure they are as aligned as possible with our goals.”

- p.7 It is not clear what the "echoes" analysis is.

We recognise that this was poorly explained in our initial submission and thank the reviewer for making sure that this important analysis is easily understood. We have made the following alterations in both the results and in the methods:

Results:

“Following Leech and colleagues[25], we began our analysis by identifying how the timeseries of 17 well-established networks[26] are represented in left DLPFC, parcellating this region into partially overlapping sub-regions or “echoes”[25] corresponding to each network (see Methods). In the context of our experiment, these correspond to aspects of the left DLPFC in which the neural signals are correlated with signals arising from other regions of cortex. Next we produced a matrix of network interactions within DLPFC, which describes how correlated each of these signals was for each individual, allowing us to test how the functional coupling of signals from different networks predicts experience in the lab. Finally, this matrix

was analysed to examine if they predicted individual variation in patterns of off-task thought recorded outside the scanner.”

Methods:

“To understand whether the interaction of signals from the whole brain were implicated in the control of off-task thought within this region, we performed a modified version of an “echoes” analysis. In the original analysis[10], an independent component analysis was performed within a masked region of the brain, to identify different components within the region associated with each independent time series. These components represent clusters of voxels in the specific region that show variation with different functional networks. In our analysis, instead of identifying these components in a data-driven way, we used 17 well-established networks from the literature. The timecourse from these networks were correlated with the timecourse within the left DLPFC to identify components within this region that represented each network. To do this, the 17 Yeo network masks were binarised and merged into a single 4D nifti file. In order to reduce statistical bias from the region itself, the corresponding region (left DLPFC) was masked out of this nifti. These networks were entered into a dual regression that extracted the timeseries from within each Yeo network and subsequently regressed these against each subjects 4D dataset within the left DLPFC. Thus, for each network, each voxel in the left DLPFC was given a value for how much it represented the Yeo network in question, to make up 17 different components within this region that each represented one of the Yeo networks. These maps were again merged into a single file and entered into the first step of a second dual regression in order to extract the timeseries of each Yeo 17 “echoes” or component.”

- p. 9 The segregation analysis is also a bit unclear. First of all, this relies strongly on the premise that whatever differences are measured between the two tasks are a stable individual difference, not dependent on circumstances, since you relate it to differences in brain structure, which obviously only show a person's long-term traits.

We thank the reviewer for highlighting this issue. In terms of whether the measurement derived from experience sampling can be considered a trait, in this study we sampled an individual's experience across three days in the laboratory, and in one session in the scanner. The metrics generated in this manner were correlated (see Methods). Together we believe that these aspects of our measurements suggest that they reflect relatively stable individual differences. Second, we note that the assumption that variance associated with a trait is a common approach within the psychological literature (for example see McVay and Kane, 2009). We have added information about this in the results:

“Unlike Experiment 1, this analysis examines off-task thinking from the perspective of a trait (see: [8-10, 23, 24] for prior examples of such an approach). Accordingly, our behavioural sessions in Experiment 2 took place across three separate days to maximise the chances that our MDES captured a reasonably consistent description of an individual's patterns of experience.”

Secondly, it is not quite clear why you expect the observed differences between the dorsal and ventral DLPFC. It would be helpful to explain the logic here.

Thanks for highlighting this lack of clarity in our initial submission. We have attempted to reduce the ambiguity for the reader in this revision by stating:

“Our functional analysis indicated signals arising from DAN had a complex topographic pattern within DLPFC, with positive coupling within a dorsal region (BA 9) and negative coupling in a ventral region (BA 46, Figure 4). This separated the region along the border of a sulcus, with the more dorsal region coupled positively to signals related to the task, and the more ventral portion related negatively to the same signals. We hypothesised that if these regions play an important functional role in how individuals focus on self-generated information, then increasing off-task thinking in the 0-back task should be linked to relatively less cortical thickness in regions of left DLPFC sensitive to signals from the DAN.”

Finally, again the pie charts in Fig 4 are unclear.

We have added the following sentence to clarify the meaning of the pie charts:

“The amount of overlap with each parcel of the Glasser parcellation is shown by the pie charts, both for the region as a whole and for each sub-region (dorsal: red box, ventral: blue box).”

- p.10 Refers to a debate in the mind-wandering literature--it would be helpful to identify this debate.

Thank you for highlighting this lack of clarity. As part of our extended introduction, we have outlined this debate and have made the reference to it in the discussion more explicit. We hope that these changes help embed our results within this theoretical context.

- p.10 Given the considerations above, I would phrase my conclusion less strongly: the DLPFC may help to prioritise cognition

We agree that this is an important issue to have clarity on and have amended our statement accordingly:

“Together these provide converging support for the involvement of DLPFC in the process of prioritising cognition that matches the demands of a particular context.”

- p.18 "This revealed a region of posterior cingulate cortex whose neural signature was consistent with a pattern of context regulation for the level of detail in ongoing experience" What is context regulation? How is this pattern consistent with that?

We thank the reviewer for highlighting the ambiguity on this issue in the initial submission. We have added an explanation of context regulation to the extended introduction and also made the following alteration to the results:

“This revealed a region of posterior cingulate cortex whose neural signature was consistent with a task-dependent association with the level of detail in ongoing experience, in a similar way to the DLPFC was related to off-task thought (see Supplementary Figure 5).”

- p.19 Several null relationships are mentioned, but to be sure there are really no relationships (and this is not an artifact of data that are too noisy) you need to either compute Bayes Factors or perform equivalence analyses.

We thank the reviewer for pointing us towards this means of further understanding the relationships in our experiment. Following this suggestion we have performed equivalence analysis following the recommendations of Lakens (2017). We would like to particularly thank the reviewer for this suggestion as it has helped quantify aspects of our results in a manner which has substantially improved the quality and hopefully impact of our work.

Results:

“To address the selectivity of the association between neural process in the DLPFC and on-task thought, we performed a number of post hoc analyses. First, using the data we collected in Experiment 1 we extracted the relationship between brain activity in the same area of DLPFC and the other components of thought (Detail, Modality, and Emotion) to see if this region played a role in task-dependent regulation of these. We subtracted the relationship to each component in the 0-back from that in the 1-back, and used the effect seen for off-task thought in Experiment 1 (Cohen’s $d=0.4777$) to define the size of the effect we were interested in. We performed equivalence tests[27] to see if the relationship between the task and thoughts for any other component could be dismissed as null. These were all

significant (Detail: $t(59)=2.000$, $p=.025$; Modality: $t(59)=2.628$, $p=.005$; Emotion: $t(59)=3.311$, $p=.001$), suggesting these effects are equivalent to zero and can be rejected as null effects (see Supplementary Figure 5). We were significantly powered to perform this analysis (recommended $n=38$). This analysis indicates that task relevant differences in the association with experience were only significant within the left DLPFC for the off-task component. We also performed an equivalence analysis that examined how unique the associations are between cortex-wide signals and patterns of experience within left DLPFC that was observed in Experiment 2 (see Supplementary Table 3). We were significantly powered to perform this analysis (recommended $n=100$). In brief, this found that no other pattern of experience could be predicted based on interactions between the same pair of networks (posterior DAN and lateral DMN) assuming we were looking for an effect of a statistically equivalent size to our significant finding. Moreover, of all the other network pairs included in our analyses all but one association with experience failed to pass Bonferroni correction for the number of comparisons. The outstanding pattern indicated associations between a different pair of networks (network 10, anterior limbic; network 16 - DMN core) within DLPFC was related to thoughts. Coupling between signals from these networks was associated with levels of detail in the 1-back task ($F(1,135)=12.792$, $p=.0005$). The same equivalence analysis for this effect showed that the relationship between this interaction and detail in the 0-back was potentially of a comparable size and so could not be dismissed as a null finding. Additionally, the effect of this interaction on off-task thought in the DLPFC in both tasks was also too large to dismiss as definitely null. Post-hoc analysis showed that the correlation between these network components in DLPFC was positively related to detailed thought in the 0-back and negatively related to it in the 1-back (see Supplementary Figure 7). Similarly, this same interaction was related negatively to off-task thought in the 0-back and positively in the 1-back. This suggests that while this region was not identified as related to detail during task performance, there may be signals in this region that also describe the task relevant moderation of levels of detail, and it cannot be ruled out that these same signals relate to off-task thought. Third, we repeated this analyses using bilateral parietal regions linked to on-task thought in Experiment 1 as the ROIs (see Methods for explanation, Figure 2 for the

regions-of-interest, and Supplementary Table 3 for equivalence results). This found no comparable evidence that integration of distributed neural signals in these regions of parietal cortex are linked to patterns of experience. Finally, we repeated the whole brain analysis from Experiment 1 looking at the neural correlates of the other components of experience identified by PCA. This revealed one significant effect: reports of detailed thought was positively associated with neural signals in the posterior cingulate cortex in the harder 1-back task than in the easier 0-back task (See supplementary Figure 6). Taken together these supplementary analyses show that (i) off-task thought was the only pattern of experience that was associated with clear task differences in its association with neural activity in the left DLPFC during task performance (Experiment 1) and (ii) the association between signals from the posterior DAN and the lateral DMN within DLPFC only are specifically related to the prioritisation of personally relevant information when external demands are reduced (Experiment 2).”

Methods:

“Several analyses were performed to assess the specificity of this result. First, we repeated this masked network interaction analysis with the other components of thought (Detail, Modality, and Emotion: see Supplementary Figure 1), and found only one significant effect within the model for Detailed thought that passed Bonferroni correction, with the interaction between Yeo networks 10 and 16 within the DLPFC region-of-interest significantly related to the level of Detail in participants’ thoughts in the 1-back task ($F(1,135)=12.792$, $p=.000484$). Next, we repeated the analysis from Experiment 1 using the other components in turn (i.e. the PCA scores from Detail, Modality, and Emotion rather than Off-task). This revealed a region of posterior cingulate cortex whose neural signature was consistent with a task-dependent association with the level of detail in ongoing experience, in a similar way to the DLPFC was related to off-task thought (see Supplementary Figure 6). To see whether it regulated detail using a similar mechanism, we repeated the masked connectivity analysis described above on this region and there was no individual interaction that passed Bonferroni correction. No other components of experience

showed a brain region with a pattern consistent with context regulation, but there were several results related to levels of thought in a non-task-dependent manner (see Supplementary Figure 9). Third, we repeated the connectivity analyses from Experiment 2 using the whole brain, rather than limiting the analysis to the DLPFC. This analysis found no significant relationships between network interactions and off-task thought in any conditions, suggesting the association between the interaction of the dorsal attention network and the default mode network with off-task thought is not a property of the broader cortical mantle. Fourth, we explored whether this same analysis would identify similar effects in regions important for on-task thought regardless of the task (lateral parietal regions: see Figure 2). This analysis revealed no significant relationships that passed Bonferroni correction suggesting that the relationships between network interactions in DLPFC and off-task thought identified in our prior analyses were unique to regions that responded in a manner that was task-dependent.

Finally, to test whether the specific relationships we found (between the network 5-network 17 interaction and off-task thought, and the network 10-network 16 interaction and detailed thought) were truly unique to these conditions, we performed a series of equivalence analyses. These were done using the TOST equivalence test for correlations described in Lakens (2017)[27]. We extracted the strength of the relationship in our significant result (between the interaction of network 5 and 17 and off-task thoughts in the 0-back, and between the interaction of network 10 and 16 and detailed thought in the 1-back) and used these as the upper and lower bounds to see if there were any effects of this size under the other conditions. All of these tests using the network 5-network 17 interaction were significant, suggesting the correlations between this interaction and the thoughts under other conditions were equivalent to 0 and represent true null findings (see Supplementary Table 3). This suggests that this interaction is specifically able to predict off-task thoughts in the 0-back task within DLPFC. The network 10-network 16 interaction was able to significantly predict detail within the DLPFC in the 1-back task. An equivalence analysis suggested that the effect in the 0-back was of equivalent size and cannot be dismissed as a null effect. Interestingly, this also involves the default mode network, and a post hoc analysis showed that the

relationship to detail was task-dependent (see Supplementary Figure 7). The relationship between this interaction and off-task thought also was not significant using this equivalence test, so this relationship cannot be dismissed as null. Post-hoc analyses showed that this relationship was also task-dependent (see Supplementary Figure 7).”

Minor points:

- p.2 suggested insertion: "We established neural activity in DLPFC is high BOTH when 'on-task' under demanding conditions and 'off-task' in a non-demanding task"

We have made this change.

- p.2 "segregation": what is segregated from what?

We have changed the wording of this slightly, and we hope our improved explanation of the echoes analysis will clarify this point.

“show lower correlation, or separation, of neural signals”

- p.7 "Post-hoc analyses confirmed this pattern was absent from regions linked to on-task thought in Experiment 1" It would be helpful to give some examples of those regions here.

We have made the following change and pointed the reader to where they can see these regions:

“Third, we repeated this analyses using bilateral parietal regions linked to on-task thought in Experiment 1 as the ROIs (see Methods for explanation, Figure 2 for the regions-of-interest, and Supplementary Table 3 for equivalence results).”

- Fig.3 This figure would be a lot easier to understand if the Yeo et al networks were associated with names.

Thanks for this suggestion. It is difficult to fit the full names of the networks onto the figure. To make sure that readers have the information available them if needed, we have included a supplementary Figure (8) in which the network names are linked to the network numbers.

Figure S 8: Full description of the Yeo 17 networks. Names are taken from Baker, J. T., Holmes, A. J., Masters, G. A., Yeo, B. T., Krienen, F., Buckner, R. L., & Ongür, D. (2014). Disruption of cortical association networks in schizophrenia and psychotic bipolar disorder. *JAMA psychiatry*, 71(2), 109-118.

- p.10 "Our individual difference analysis suggests DLPFC helps prioritise off-task thought via reductions in the processing of external task-relevant signals[3]," I find this very helpful. It would have been nice to explain this reasoning in the results already so it is clearer why the connectivity analyses were done.

We thank the reviewer for pointing out this instance as an example to use in our explanations. We have made the following change:

“Extrapolating from these accounts, we hypothesised that context-dependent prioritising of off-task thought observed in our prior analysis occurs because of how neural signals related to the external task (i.e. posterior elements of the DAN) are processed in left DLPFC.”

- p. 16 "eleven participants excluded in total." seems to be an orphan sentence. Something like "leading to eleven participants being excluded in total." would make more sense.

We have made this change.

- p.16 "The PCA loadings for the equivalent components were correlated across experimental conditions" It would be helpful to mention the actual correlation coefficients here.

In this revision we have included the relevant correlations.

- p.17 "with a time period of 6 seconds prior to the MDES probe and the scores for the task-related component" Did you take into account the delay in the HRF?

We have added a sentence to clarify that there was convolution with the HRF.

- p.17 The Yeo analysis is very hard to understand. Maybe it would be clearer if the purpose of each analysis step could be explained

We hope that this analysis has been made more clear by our new explanation (see point on echoes analysis above).

- p.18 What is "voxel-wise correction at $p < .05/17$ "?

We have added this explanation:

“We inspected each component, and if no voxels were significantly related to the Yeo network in question it was deemed to likely represent noise and its interactions were excluded from the analysis. An alpha value of $p < .05/17$ was used to account for family wise error in these analyses.”

What is the manova in the same paragraph trying to predict?

We have added the following clarification:

“This was used to identify any specific network interactions that could be predicted by the thoughts in either task.”

- p. 18 "we repeated this masked network interaction analysis with the other components of thought" Would be nice to give examples of these other components of thought.

We have included a list of these components and pointed the reader to the Supplementary figures where they are described in full.

- p.19 "Chord diagrams were made using R[40]." It would be nice to give some more explanation of the statistics underlying these graphs.

We have added the following explanation:

“These represent the strength of the relationship between each interaction and the thoughts. Parameter estimates were extracted from the MANOVA so that each interaction had a beta-weight representing how strongly it related to the thoughts as part of the model. These were used to create chord diagrams that show the strength

of these relationships (in the size of the chords) and their direction (blue is negative, red is positive).”

- Several references are missing page numbers and issue numbers. There is some inconsistency in the naming of journal names (such as PNAS).

Thank you for your detailed reading, we have addressed these issues and corrected any inconsistencies.

Reviewer #2 (Remarks to the Author):

The paper presents very interesting findings about the DLPFC and its function in prioritizes goal representations in an adaptive, context-dependent manner. The two experiments presented are an impressive body of work and I think the paper could have a positive and meaningful impact on the field. However, I do have a few concerns that I think should be addressed before recommending publication.

A prominent issue is the lack of an Introduction that integrates related prior work and theoretical positions. This is critical because the paper presents results on relatively specific hypothesis, so it's important to understand where it came from and what alternative explanations may exist. A more thorough explanation of why they chose to focus on the left DLPFC would be especially helpful -- theoretically as well as related to previous research. Including this will assist readers/reviewers situate and evaluate the paper. (As an aside, I noticed that most of the articles published in Nature Communications do include short introductions.)

Another issue in general is the connection among the different analyses adopted in the paper. They are undoubtedly important, novel, and interesting. However, I think more could be said in the paper about how they naturally flow together, and what question each answers. An Introduction may also help alleviate this issue.

Some of the decisions that were made/analyses completed along the way are not described is a weakness of the paper. For example, there is no mention of why the authors chose to focus on the left DLPFC in particular (rather than bilateral or right)? Why was window of 6 seconds chosen to analyze prior to the MDES questions? What is the significance/relationship of having the laboratory task as well as the scanner? Some of these questions and decisions likely have very good rationales, but they are left out of the paper. I realize there are significant space constraints, but I think some the information is necessary to include.

We thank the reviewer for highlighting these important points. Firstly, we have modified the manuscript to include a more thorough introduction, and we hope that this alleviates some of the issues mentioned by the reviews, as was suggested. In terms of these specific points, we have addressed them in order:

The left DLPFC was not chosen as a focus for this paper in a hypothesis driven manner, and we apologise for not making this clearer. While we did hope to identify a region of the brain that was involved in off-task thought during the easy task and on-task thought during the hard task, our motivation were psychological accounts that emphasise a common role of control in both task focused and internally generated cognition (i.e. Smallwood, 2013, Psychological Bulletin). Although we broadly expected this common relationship to emerge in regions linked to cognitive control, we had no prior predictions for where this might be. We have included the following sentence to clarify this:

“We performed a whole-brain fMRI analysis to see whether any regions of the brain had this neural profile.”

We chose 6 seconds because it was the longest temporal interval within which no behavioural response occurred. This was built into our design in a priori manner, so was not an analytic choice. We have now clarified this in the methods:

“We chose to use 6 seconds as it was the longest temporal interval within which no behavioural response occurred.”

The discussion section is also very short. Although there are some summary statements in the paper, but I think it might be helpful to again link the findings to a broader context and describing how each main finding is important. Sometimes I was left wondering what questions a particular analysis answered, and how that fit with current theories/findings.

It would be helpful if the discussion section broadened a bit by exploring alternative explanations as well. How do you know if it is an actual prioritization mechanism, compared to another explanation? What other explanations may exist?

Thanks for this suggestion. We have broadened the discussion to address the theoretical implications of our finding.

Why is the family wise error rate .05 (Fig 1)? In general, how were multiple comparisons dealt with here?

The group level task-based analysis was corrected for multiple comparisons emerging from the number of voxels within our matrix. In this study we adhered to best practice (i.e. Eklund et al., 2017) - We used FLAME as this has the most conservative estimates of the tests examined by Eklund and colleagues, used a cluster forming threshold of $Z > 3.1$ and finally controlled for family wise error at $p < .05$. To make this clear we have added the following sentence:

“In these analyses we followed best practice as described by Eklund and colleagues (2017). Specifically, we used FLAME, as implemented in FSL, applied a cluster forming threshold of $z = 3.1$, and corrected these at $p < .05$ (corrected for family wise error rate using random field theory).”

Although I think the Figures are very well-done and helpful, they could still use some

improvement to assist with reader interpretation by including more labels and details in the figure and captions.

There may be a better name for the section “classification of off-task thought component.” The classification part in particular was a bit confusing at first.

We have made the following alteration so as not to confuse the reader with classification in a statistical sense, and thank you for pointing this out:

“Identification of an off-task thought component”

The writing quality in the Methods section could improve quite a bit. The other sections of the paper are written much more clearly/concisely in comparison.

Thanks for this suggestion we have thoroughly proof read the Methods and Supplementary sections of this revision.

The human subjects section is somewhat difficult to understand in terms of which participants came from what group/study. Perhaps a table would help display this information.

Thank you for this suggestion, we have included a table to explain the final participants used for each experimental analysis.

Experiment	Task-based fMRI	Resting state fMRI	Cortical thickness MRI
Number of participants	60	146	142
Age (years)	M=20.21, S.D.=2.49	M=20.21, S.D.=2.49	M=2.23, S.D.=2.47
Gender	37 F, 23 M	89 F, 57 M	86 F, 56 M

Table 1. Participant demographics for each experiment. 39 participants performed both the resting state and task-based portions of this study. Cortical thickness was performed in the same group as the resting state, but four participants were excluded as their structural data did not pass quality control.

It would also be helpful to detail the number of instances for each measure (e.g. MDES in the scanner, lab, etc.). How many of these instances were used to create the PCA.

Descriptive statistics may also be useful (in supplementary materials).

This section was included in the methods to inform the reader of the number of thought probes, we have also included a sentence in the results to clarify this:

“In the online task-based fMRI part of this study (Experiment 1), participants completed this task while undergoing fMRI scanning. Each run was 9-minutes in length and there were four runs per scanning session. In each run, there was an average of six thought probes (three in each condition), so that there were on average 24 (SD=3.30, mean=12 in each condition) MDES probes in each session. Two participants had one run dropped due to technical issues, leaving them with 18 MDES probes each.

In the behavioural laboratory (Experiment 2), to derive a reasonable stable estimate of each individual’s patterns of thought, participants performed the task on three separate days in sessions that lasted around 25 minutes. In each session, there were eight blocks. In total, an average of 30.7 MDES probes occurred (SD=5.7, mean=15.4 in each condition). In the laboratory, accuracy was significantly greater ($t(145)=9.487$, $p<.001$) and reaction time significantly faster ($t(145)=14.362$, $p<.001$) in the easier 0-back task. This effect was not found in either measure during fMRI scanning (accuracy: $t(59)=0.369$, $p=.714$, rt: $t(59)=0.052$, $p=.958$, see Supplementary Figure 1).”

“In this study there were 24 MDES probes in the scanning experiment, yielding a total of 1438 observations for Experiment 1, and 30.7 on average in each session in the behavioural laboratory, yielding a total of 4482 observations in Experiment 2.”

“Two participants were excluded for falling asleep” appears in more than one section. Are these the same two people?

We thank the reviewer for pointing out this repetition. This is a mistake by the author, there were only two participants excluded for falling asleep during the task-based portion of this study. There were in fact 11 participants who had more than 15% of their data affected by motion. We apologise for this mistake and thank the reviewer for pointing this out.

Reviewer #3 (Remarks to the Author):

This is an exciting and potentially paradigm-shifting study on the brain basis of mind wandering. The experiment was cleverly-designed and well-executed; the analyses were rigorous and innovative, leveraging multiple cutting-edge approaches to fMRI data analysis to provide converging evidence in support of conclusions. The findings have important implications not only for research on mind wandering but also for other modes of self-generated thought as well as more general research on human brain networks and cognitive neuroscience.

I am very familiar with the work of this research group. In my view this study is among the most (if not the most) important to come from their lab. I do not recall ever recommending acceptance of a paper without any changes, but I could not detect anything that should be changed about this paper, aside from the minor point below. I would like to congratulate the authors on an outstanding contribution to science.

Thank you for such a positive view on our work!

Minor point

In the first sentence of the Discussion (line 2), I believe the authors meant to include the word "relevant" between "personally" and "information."

Thank you for your helpful comments, we have made this change.

Reviewers' Comments:

Reviewer #1:

Remarks to the Author:

The paper has improved a lot. Thank you for taking on board my suggestions! I have only a few minor comments remaining:

- p.7 It is not so clear what the purpose is of the intrinsic connectivity analysis of the DLPFC with other regions. What does it mean that you identify the ventral attention network?
- p.10 "Following Leech and colleagues[25], we began our analysis by identifying.." At this point, it is not clear what the point is of this echoes analysis. What question are you trying to answer with that analysis? Maybe something like how the DLPFC interacts with the rest of the brain.
- In the description of the equivalence analysis, it will be a lot clearer if you indicate that when tests are neither significant in the equivalence test nor in the original test, then the data are too uncertain to draw any conclusion.

Reviewer #2:

Remarks to the Author:

This is my second time reviewing this manuscript. As noted in my first review, the work presented here is very interesting and novel. I think it will make an important contribution to the field.

The manuscript has undergone significant revisions based on all of the concerns raised in the first round of reviews, which have substantially improved the paper.

I recommend this article be accepted for publication, with two remaining comments that can be easily addressed:

1. Although an introduction has been added in the revision, I still think it could be fleshed out a bit more. Despite a large body of previous work that could have motivated the research, the Intro/motivating review portion is limited to a single paragraph. I think more can be said here to set up the importance with additional background information, potentially competing viewpoints, and perhaps some predictions about why certain regions (e.g., DLPFC) may play an important role in off-task thought.
2. Minor: Overall, I think the figures have much improved. There is still some ambiguity that could be improved, however. For example, why is there a box around the lateral default mode column in Fig. 4. As well, I acknowledge it might just be me, but I do not quite understand the pie charts and the corresponding captions in Fig 2 -- i.e. the overlap of regions with Brodmann areas. Can the authors say more about the significance/meaning of of this calculation?

Reviewer #3:

None

Response to reviewers: Turnbull and colleagues

Reviewers' comments:

Reviewer #1 (Remarks to the Author):

The paper has improved a lot. Thank you for taking on board my suggestions! I have only a few minor comments remaining:

- p.7 It is not so clear what the purpose is of the intrinsic connectivity analysis of the DLPFC with other regions. What does it mean that you identify the ventral attention network?

Thank you for your comment, we have added the following explanations to clarify the reason for this analysis and the meaning of the results:

“To understand how this region fits into the broader neural architecture, we performed a seed-based functional connectivity analysis.”

“which has been shown to play a role in task-set maintenance[20, attentional re-orienting, and contextual cueing[21].”

“This network shows activity during spatial-orienting of visual attention and exerts top-down control over visual areas[21].”

Additionally, as requested by reviewer 2, we have included a more thorough elaboration of the ventral attention network and its functions in the introduction.

- p.10 "Following Leech and colleagues[25], we began our analysis by identifying.." At this point, it is not clear what the point is of this echoes analysis. What question are you trying to answer with that analysis? Maybe something like how the DLPFC interacts with the rest of the brain.

We thank the reviewer for pointing out this lack of clarification. We have added the following sentence to explain our motivation:

“The pattern of association between activity in DLPFC and patterns of on-task/off-task thought observed in Experiment 1 could indicate that neural signals that reflect both task-related and self-generated information are processed within this region of cortex. To test this possibility in Experiment 2 we performed an analysis to determine (a) whether neural signals arising from other regions of cortex are observed in the DLPFC and (b) if the interaction between these signals explained population variation in context regulation.”

- In the description of the equivalence analysis, it will be a lot clearer if you indicate that when tests are neither significant in the equivalence test nor in the original test, then the data are too uncertain to draw any conclusion.

Thank you for this suggestion, we have included the following sentence to improve clarity:

“This means that the relationship between detailed thought in the 0-back, and task-related thought in both tasks, and the interaction between network 10 and network 16 within DLPFC was not statistically significant but was not significantly equivalent to 0, suggesting these relationships are too uncertain to draw firm conclusions.”

Reviewer #2 (Remarks to the Author):

This is my second time reviewing this manuscript. As noted in my first review, the work presented here is very interesting and novel. I think it will make an important contribution to the field.

The manuscript has undergone significant revisions based on all of the concerns raised in the first round of reviews, which have substantially improved the paper.

I recommend this article be accepted for publication, with two remaining comments that can be easily addressed:

1. Although an introduction has been added in the revision, I still think it could be fleshed out a bit more. Despite a large body of previous work that could have motivated the research, the Intro/motivating review portion is limited to a single paragraph. I think more can be said here to set up the importance with additional background information, potentially competing viewpoints, and perhaps some predictions about why certain regions (e.g., DLPFC) may play an important role in off-task thought.

Thank you for your comment. We have broadened the introduction to include more information about the ventral attention/saliency network and why we might expect regions within it to be involved in this process. We feel that it would be inappropriate to develop competing viewpoints regarding specific brain regions, as when we conducted this study we did not have any specific regional predictions and did not want to run the risk of hypothesising after the fact (i.e. Kerr, N. L. (1998). HARKing: Hypothesizing after the results are known. *Personality and Social Psychology Review*, 2(3), 196-217).

2. Minor: Overall, I think the figures have much improved. There is still some ambiguity that could be improved, however. For example, why is there a box around the lateral default mode column in Fig. 4. As well, I acknowledge it might just be me, but I do not quite understand the pie charts and the corresponding captions in Fig 2 -- i.e. the overlap of regions with Brodmann areas. Can the authors say more about the significance/meaning of of this calculation?

Thank you for pointing out the lack of clarity in Figure 4. In this figure we used colour to help the reader see the conceptual links across the different aspects of this figure. We hoped that the red box around the pie chart and the red dashed line around the dorsal aspect of DLPFC would help the reader see that these aspects of the figure are related. To reduce ambiguity in this revision we have also annotated the pie charts with appropriate labels. The box around the lateral default mode column was included to show how this network was also represented within this region, but make it clear to the reader that the cortical thickness analysis was performed using the posterior dorsal attention network (i.e. all of the information outside the box was used for the same analysis). The pie charts represent the overlap with the Brodmann atlas, which will help the reader locate the region in a well defined anatomical context. The DLPFC result is mostly located in region BA9 and partially in BA8 and BA46, whereas the on-task related regions are mostly in BA7 and partly in BA19.

Reviewers' Comments:

Reviewer #2:

Remarks to the Author:

The authors have addressed the reviewers' remaining concerns. I recommend acceptance for publication at this time. Congratulations on a great piece of work.